# Modelling the Influence of Waste Rubber on Compressive Strength of Concrete by Artificial Neural Networks

**DOI:** 10.3390/ma12040561

**Published:** 2019-02-13

**Authors:** Marijana Hadzima-Nyarko, Emmanuel Karlo Nyarko, Naida Ademović, Ivana Miličević, Tanja Kalman Šipoš

**Affiliations:** 1Faculty of Civil Engineering and Architecture, University of J.J. Strossmayer, Vladimira Preloga 3, 31000 Osijek, Croatia; ivana.milicevic@gfos.hr (I.M.); tkalman@gfos.hr (T.K.Š.); 2Faculty of Electrical Engineering, Computer Science and Information Technology Osijek, Kneza Trpimira 2b, 31000 Osijek, Croatia; nyarko@ferit.hr; 3Faculty of Civil Engineering in Sarajevo, University of Sarajevo, Patriotske lige 30, 71000 Sarajevo, Bosnia and Herzegovina; naidadem@yahoo.com

**Keywords:** tire rubber concrete, compressive strength, artificial neural networks, database of experimental results

## Abstract

One of the major causes of ecological and environmental problems comes from the enormous number of discarded waste tires, which is directly connected to the exponential growth of the world’s population. In this paper, previous works carried out on the effects of partial or full replacement of aggregate in concrete with waste rubber on some properties of concrete were investigated. A database containing 457 mixtures with partial or full replacement of natural aggregate with waste rubber in concrete provided by different researchers was formed. This database served as the basis for investigating the influence of partial or full replacement of natural aggregate with waste rubber in concrete on compressive strength. With the aid of the database, the possibility of achieving reliable prediction of the compressive strength of concrete with tire rubber is explored using neural network modelling.

## 1. Introduction

Globally, 1.5 billion tires are produced annually, of which 300 million are produced in the USA [1]. It is believed that all of these tires will become waste tires. In the process of waste disposal/management, the reduction of waste and its recycling are very important features, both from an economical and environmental perspective. Tire waste is an important and quantitatively significant component of total waste composition. Therefore, it is extremely important to pay attention to the recycling and reuse of tire waste in order to conserve natural resources and reduce the landfill needed for its disposal. The time required for waste tire rubber decomposition is greater than 50 years. The quantity of discharged tires is increasing very quickly, since the usage of tires increases on a daily basis.

Waste tires represent a potential for re-entry into the market through new materials and new products. Recently, tire waste found its use in the cement industry as a substitute source for the production of playground matt, and for various coverings. This material found its application in various fields, from different kinds of barriers and posts, as well as in asphalt pavement mixtures [2]. Governments of certain countries (USA and France) have made the use of crumb rubber in highway construction mandatory for projects under their finance [3]. Savas et al. [4], Benazzouk and Queneudec [5], and Paine et al. [6] investigated the influence of rubber addition to concrete mixes on freezing and thawing resistance. More recently, the use of waste tires and its components in the technology of concrete production has been investigated.

The long lifetime of waste tire rubber was one of the factors which aroused the interest of the research community in the replacement of natural river aggregate with rubber products in concrete mixtures. This was with the goal of producing environmentally-friendly concrete [7]. The ductility of concrete can be improved by substituting natural aggregates with various forms of recycled tire rubber—powder, crumb or chipped rubber particles—thereby taking into consideration sustainability and environmental pollution [8]. 

Good characteristics of recycled tire rubber, low elastic modulus and highly ductile-engineered cementitious composites repair material were used to alleviate repair failure induced by restrained drying shrinkage [9]. Higher contents of waste tire crumb rubber particles used in concrete produces lightweight concrete and increases the workability of concrete [10]. By replacing natural aggregate with rubber waste of various percentages and making a comparison with regular concrete, it was noted that there is a reduction in the static and dynamic moduli of elasticity in the samples with rubber waste additives [11].

Rubber aggregates in concrete have the greatest impact on concrete compressive strength [12]. El-Gammal et al. [13] provided results on the compressive strength of concrete with 100% replacement of chipped rubber, which was reduced by 90% with respect to the original mixture. In the case of complete replacement of sand with crumb rubber, the reduction in strength was as high as 80%. The reduction in compressive, splitting tensile and flexural strength was reported by Panda et al. [14]. When 15% of the coarse aggregate was replaced with rubber particles, strength decreased by 45%. Reductions in the unit weight of concrete depends on the type and the content of crumb rubber (10–30%) and is in the range of 14% to 28%.

In recent years, interest and application of artificial neural networks (ANNs) has increased due to their simplicity and efficiency in creating input-output black box models. ANNs have been applied in diverse areas such as medicine, business, physics, geology, engineering, and environmental engineering in particular [15,16,17]. 

Although the prediction of concrete properties with natural aggregates using artificial neural network (ANN) modeling has been widely researched [18,19,20,21], few studies involve modelling the compressive strength of concrete comprising of waste tires. Topcu and Sarıdemir [22] investigated the properties of fresh concrete using artificial neural network (ANN) and fuzzy logic (FL). They used 36 experimental results with their ANN and FL models. They demonstrated that by using ANN and FL models, it is possible to determine the properties of fresh concrete without conducting any experiments. This was based on comparing experimental results with those of ANN and FL models.

Prediction of the compressive strength of rubberized concrete based on test measurements by utilizing an ANN was reported by Abdollahzade et al. [23]. Compared to a multiple linear regression (MLR) model, the back-propagation ANN model was able to predict the strength of rubberized concrete with a suitable degree of accuracy.

Application of an ANN was reported by Diaconescu et al. [24] in the domain of filler and resin content influence on the mechanical properties of polymer concrete with powdered tire waste. The ANN was used in the process of optimization of the filler material, made of mixed epoxy resin, aggregates and tire powder. Results of the modelling was a mixture of maximum strength and lowest cost. 

Gesoglu et al. [25] proposed ANN- and genetic programming (GEP)-based explicit models for the prediction of mechanical properties (compressive strength, splitting tensile strength, and static elastic modulus) of rubberized concretes based on an experimental study containing 70 rubberized concretes. In order to formulate their model, one output and eight design variables were used. A high forecast capability with a given accuracy was obtained by both methods.

From the literature review, it can be stated that not a lot of research has been carried out using ANNs to predict the compressive strength of concrete with tire rubber. None of them involved a large enough database needed to attain a more reliable estimation of compressive strength. The first contribution of this study is to compile a suitably large database with enough experimental results (457 experimental tests of compressive strength of concrete with tire rubber). 

Irrespective of the experimental studies on concrete with tire rubber, no algorithm exists for defining concrete compressive strength. The second contribution of this study is to use the data in the generated database to create an ANN model capable of predicting compressive strength to a suitable degree of accuracy.

## 2. Database Description

A methodical search of published papers, using sources such as Springer, Scopus and Google, dealing with cement-based rubber concrete properties up to January 2018 was conducted. To focus on the study of the compressive strength of rubber concrete, rubber mixtures such as asphalt rubber concrete or rubber mortar were not considered. In addition, publications written in a non-English language, or ones of local rather than international affiliation, were also excluded. To ensure that there were no omissions, broad keywords such as concrete, rubber, rubber concrete, rubberized concrete, and rubcrete were used.

The collected experimental database contains data from 457 published tests (Table 1). Selection of the database parameters was done based on the information available for data samples from all tests. If some parameters were not available for the published tests, those test data were not included in the database.

Among the first studies considered was that from Toutanji in 1996 [2], who used rubber tire particles to investigate the mechanical characteristics of obtained concrete. One of the conclusions from these investigations was that there was a reduction in compressive and flexural strength with the inclusion of rubber tire chips in concrete. The inclusion of rubber tire aggregate in the concrete samples revealed a ductile failure, exhibiting substantial displacements before fracture. 

More researches were provided in order to investigate mechanical properties of rubberized concrete. The largest database of performed tests was done in [26], where the experimental campaign was formulated with the goal of determining the mechanical characteristics of the rubberized concrete with and without silica fume. The idea was to replace parts of fine and coarse aggregate; in that respect, two types of tire rubber were used, crumb rubber and tire chips. The content of rubber varied from 2.5% to 50% with respect to the total aggregate volume providing six different designed rubber mixtures. As in the previous case, a significant decrease of strength and modulus of elasticity was noted with an increase in rubber percentage. On the contrary, the addition of silica fume lowered the strength loss rate and to a certain extent enhanced the mechanical properties of the rubberized concrete. 

In [27], apart from mechanical tests, selected standard durability tests were also performed. In this case, as in [2,26], the same strengths were tested. The two tests that were conducted for the investigation of durability were permeability and water absorption. It was only after increasing the replacement of aggregate by 5% of rubberized material that significant variations were noted. 

In Paine [6], the experiments revealed that the inclusion of granulated rubber may increase the freeze–thaw resistance of concrete. Additionally, the use of this type of rubber provides lower thermal conductivity of concrete which is appropriate for application in floors and foundations deprived of the requirement for matching insulation systems. 

The authors in [28] concluded that, regardless of the numerous studies which have been conducted regarding the properties of concrete with crumb rubber used as replacement for fine aggregate by volume, no guidelines exist to advice on the composition of rubbercrete. Thus, on the basis of the database they generated, they offered one such design guideline. The experiment consisted of 45 concrete mixes which were tested in fresh state. Other measurements were performed and included air content, testing various strengths (compressive, splitting tensile, flexural), testing the modulus of elasticity and slump (a mechanical property of rubbercrete).

There are, mainly, two ways of adding rubber aggregate to concrete: (i) adding rubber aggregate relative to the weight or volume of concrete directly; or (ii) substituting aggregates of concrete by weight and by volume. Due to the fact that the density of natural aggregate in concrete is approximately 2.5 times that of rubber, it means that if weight substitution is used to replace rubber in the aggregates, the volume of the resulting mixture would be much higher than the original one. From a practical viewpoint, substituting natural aggregate of concrete with rubber aggregate by volume is more suitable.

Those researches were taken into consideration in order to investigate various characteristics of concrete with rubber waste. As previously stated, in this paper, only the influence of partial or full replacement of natural aggregate with waste rubber in concrete on the compressive strength of rubberized concrete is investigated. Figure 1. shows the influence of partial or full replacement (in percentages) of aggregate with waste rubber on compressive strength. 

The first conclusion that can be obtained from Figure 1. is that the addition of waste rubber aggregate results in a decrease in the compressive strength of concrete, and this decrease becomes more significant as the proportion of rubber aggregate in the mixture increases. 

The reasoning behind this could be: Rubber aggregate is significantly softer than natural aggregate.Thus, when natural aggregate is replaced by rubber aggregate, the load carrying material in the mixture is reduced, resulting in a decrease of compressive strength.In addition, rubber particles, which are less stiff than natural aggregate, can be considered as gaps in the concrete mixture.At the boundary between the rubber particles and concrete matrix, stress concentration usually arises. Due to the elastic mismatch, during loading, cracks are initialized around the rubber aggregate and quickly develop around the rubber particles, expanding rapidly throughout the matrix causing increased rupture in the concrete.Due to their non-polar nature, air is captured on the surface of rubber particles, and causes great inconsistency between rubber aggregate and the surrounding cement.

Since, according to the Table 1, various sizes of waste rubber aggregate were included in the database, rubber aggregate was divided into two types: fine rubber (0–4 mm) and coarse rubber (4–16 mm). Each aggregate partition was also divided into two sub-parts: fine (0–4 mm) and coarse aggregate (4–16 mm). 

The following equation defines the total aggregate ratio:NA-F [%] + NA-C [%] + RB-F [%] + RB-C [%] = 100%(1)
where NA-F denotes fine natural aggregate in percentage of the total aggregate, NA-C denotes coarse natural aggregate in percentage of the total aggregate, RB-F denotes fine rubber in percentage of the total aggregate, RB-C denotes coarse rubber in percentage of the total aggregate.

The experimental database consists of six input parameters: cement (kg); w/c ratio; fine natural aggregate (NA-F) (%); coarse natural aggregate (NA-C) (%); fine rubber (RB-F) (%); and coarse rubber (RB-C) (%). For structural applications, one of the most important mechanical properties of concrete is the compressive strength. Thus, in order to evaluate the performance of aggregate substitution with rubbers, it is necessary to estimate the compressive strength of concrete at the age of 28 days (*f*_c,28_ in MPa). This performance indicator represents the output parameter. The general distribution of input and output data with the minimum, maximum and average values are presented in Table 2. 

The histogram of the percentage of RB-F compared to the fine aggregate (NA-F) is presented in Figure 2a). It can be seen that 159 examples (or 34.78% of all specimens) contain a maximum of 5% of RB-F. Most of the provided experiments (89.72% of samples) contain up to 25% of replacement of RB-F compared to the fine natural aggregate. Only 15 specimens contain up to 75% replacement of RB-F compared to the fine natural aggregate and 32 specimens contain 100% replacement of RB-F compared to the fine natural aggregate.

When coarse rubber aggregate replacement is being considered, the situation is slightly different (Figure 2b). Most of the provided experiments (92.56% of samples) contain up to 20% of replacement of RB-C compared to the coarse natural aggregate (NA-C). Eleven specimens contain 50% replacement of RB-C compared to the coarse natural aggregate and only six specimens contain 100% replacement of RB-C compared to NA-C.

The weight of the cement in the database is in the range of 250 to 650 kg (Figure 2c). Thirty-eight (38) mixtures in the database contain 300 kg of cement, 27 mixtures contain up to 325 kg, 64 mixtures contain up to 350 kg, 59 mixtures contain up to 375 kg, 85 mixtures contain up to 400 kg, 44 mixtures contain up to 425 kg, 70 mixtures contain up to 450 kg and 27 mixtures contain up to 475 kg.

The w/c ratio of the database is presented in Figure 2d. The w/c ratio is in the range of 0.25 to 0.7, with 144 mixtures having a value of 0.45. One hundred and ten (110) mixtures have a w/c ratio value of 0.6.

## 3. Neural Network Modelling

Artificial neural networks (ANNs) are often used for modelling complex nonlinear functions determined from observations. ANN has proven to be an efficient tool for engineering applications due to its powerful data processing and learning capabilities [15,16]. Therefore, in this study, ANN modeling will be performed and the resulting model used for prediction of the compressive strength of concrete containing waste tires.

### 3.1. Neural Network Architecture

The basic structure of the most popular multi-layer feedforward ANN generally consists of three distinctive layers, each of which is made up of nodes or neurons: the input layer, one or more hidden layers, and the output layer. In such a network, data only travels forward through the layers. The input layer is used for the introduction of data to the model. The hidden layers are used for data processing while the output layer is used to obtain the model output. The neurons of a given layer are linked to other neurons in the following layer. Apart from the neurons in the input layer, which only passes data through, all the other neurons are made up of several components: weights, an offset or bias and an activation function (Figure 3). Common activation functions include the linear function *purelin* (2), and non-linear sigmoid functions *logsig* (3) and *tansig* (4) which are defined as follows: y = x(2)
(3)$$y=\frac{1}{\left(1+e^{-x}\right)}$$
(4)$$y=\frac{2}{\left(1+e^{-2x}\right)}-1$$

ANN training involves determining the set of optimal values of weights. This is achieved by determining the model errors, i.e., the difference between the obtained model outputs and the measured or desired output values. The optimal values of the weights are then determined by minimizing a cost function which is basically the mean squared error (MSE). This training process is performed iteratively for a given number of cycles until a suitable network output accuracy is obtained. In this study, the Levenberg-Marquardt method is used, in combination with Bayesian regularization, to determine the optimal values of the weights. This optimization procedure minimizes a combination of squared errors and weights, and then determines the suitable combination in order to enhance the generalization properties of the network. 

In order to improve the generalization properties of ANN, the experimental data needs to be divided into three subsets: training, validation, and testing datasets. In this study, 70% of the original experimental data was randomly selected as the training dataset, 15% as the validation dataset and 15% as the test dataset. The training dataset is used to determine the model ANN weights (or parameters) i.e., for model calibration, the validation dataset is used during training to prevent overfitting, i.e., to improve the generalization capabilities of the model, while the test dataset is used to evaluate or validate the model.

The number of neurons in the hidden layers greatly affects the stability of the ANN. Randomly selecting the number of hidden neurons may cause either underfitting or overfitting of the models [70]. Underfitting of the model is achieved if the number of hidden neurons is too small, and as such the model is incapable of modelling the nonlinearity of problem. Overfitting may happen due to an excessive number of hidden neurons, and as such, the ANN model overestimates the difficulty of the given problem. Thus, it is necessary to determine the suitable number of hidden neurons in order to avoid overfitting so as to have a model that generalizes well and has the lowest possible deviation in prediction.

In this paper, the optimal number of neurons for the ANN model was determined by carrying out a 5-fold cross validation using the training dataset. Since the number of neurons in the input and output layers are determined by the number of input and output parameters respectively, cross validation was used to determine the number of neurons in the hidden layer. ANN models containing one, two and three hidden layers were analyzed. All possible combinations were considered using 2 to 20 neurons for the models with one hidden layer, 2 to 20 neurons per layer for the models with two hidden layers and 2 to 10 neurons per layer for the models with three hidden layers. A flow chart of the model selection procedure is given in Figure 4. For a given network architecture, the training procedure was repeated 10 times, with random initialization of weights each time, and the best trained network was selected as a representative of the given architecture.

Using the previously described procedure, the most suitable ANN model obtained consisted of three hidden layers with nine, three and two hidden neurons respectively (Figure 5), with six neurons in the input layer and one neuron in the output layer.

### 3.2. Performance Evaluation

In order to evaluate the effectiveness and predictive accuracy of each, three statistical performance measures were used: the coefficient of correlation (*R*), the root mean square error (*RMSE*) and mean absolute percentage error (*MAPE*), defined as follows:(5)$$R=\frac{\sum_{i=1}^{n}\left(EV_{i}-EV_{mean}\right)\cdot \left(PV_{i}-PV_{mean}\right)}{\sqrt{\sum_{i=1}^{n}{\left(EV_{i}-EV_{mean}\right)}^{2}}\cdot \sum_{i=1}^{n}{\left(EV_{i}-EV_{mean}\right)}^{2}}$$
(6)$$RMSE=\sqrt{\frac{\sum_{i=1}^{n}{\left(PV_{i}-EV_{i}\right)}^{2}}{n}}$$
(7)$$MAPE=\frac{1}{n}\sum_{i=1}^{n}\left|\frac{EV_{i}-PV_{i}}{EV_{i}}\right|$$
where *EV*_i_ is the measured or experimental value of compressive strength, *PV*_i_ is the predicted value or model output (compressive strength), *EV*_mean_ is the average measured or experimental value of compressive strength, *PV*_mean_ is the average model output value representing the compressive strength and *n* is the number of data samples.

Efficient models with good predictive capabilities have lower *RMSE* and *MAPE* values and higher R values.

## 4. Results and Discussion

Three suitable ANN models were determined having one, two and three hidden layers using the procedure outlined in the previous chapter and shown in Figure 4. Parameters of the proposed models were determined using the train dataset while the obtained models were evaluated using the test dataset. *RMSE* was used as the cost function during model training, while *R* and *MAPE* values were used to evaluate the obtained models.

For the ANN model with a single hidden layer, the best results were obtained with 10 neurons in the hidden layer (Figure 6a). Similarly, the best ANN model with two hidden layers was obtained with a structure having seven neurons in the first hidden layer and two neurons in the second hidden layer (Figure 6b). For the ANN model with three hidden layers, the best model had nine neurons in the first hidden layer, three neurons in the second hidden layer and two neurons in the third hidden layer (Figure 6c).

For the ANN model with one hidden layer and 10 neurons in the hidden layer, the performance criteria expressed in terms of the values of *R* and *RMSE* for the train and test dataset are presented in Figure 7 and Figure 8.

The results for the three ANN models are also presented in Table 3. Since the selection of a suitable model is made based only upon the information obtained during the training phase, it can be concluded that the ANN model with three hidden layers is the best since it has the highest value of *R* and the lowest values of *RMSE* and *MAPE* on the train dataset. The next best model is the ANN model with one hidden layer followed by the ANN model with two hidden layers. The order of the quality of the ANN models is further confirmed by the results obtained by the models on the test dataset where the best results are as well obtained using the ANN model with three hidden layers. 

The percentage error for the train and test datasets for the best ANN model with three hidden layers are displayed in Figure 9. It can be seen that maximum absolute error obtained on the train data is about 20, while a maximum value of about 26 is obtained on test data. 

## 5. Conclusions

A database containing 457 mixtures with partial or full replacement of natural aggregate with waste rubber in concrete provided by different researchers was formed. This database served as the basis for neural network modelling in order to achieve a suitably accurate prediction of the compressive strength of concrete with tire rubber. Artificial neutral networks have shown to be able predict results from examples and observations. This makes them a powerful tool for solving complex problems in various fields and specifically in civil engineering. Various artificial neural network models for determining the compressive strength of concrete at the age of 28 days with tire rubber were considered. Models having one, two and three hidden layers and various numbers of neurons were analyzed. The model with three hidden layers, with nine, three and two neurons respectively, gave the best results with respect to prediction accuracy. This model had the highest *R* value of 0.96 and 0.98 for the train and test data, respectively, an achieved the lowest *RMSE* and *MAPE* values (4.8 and 20.2 for the train data, respectively, and 3.78 and 21.6 for the test data, respectively). The maximum absolute error obtained on the train data is about 20, while a maximum value of about 26 was obtained on test data. The results were obtained without any additional experiments in a fairly short period of time and with relatively acceptable errors. The application of an ANN multilayer backpropagation network in the field of predicting the compressive strength of concrete with tire rubber is appropriate and can be seen as an alternative and suitable approach.

## Figures and Tables

**Figure 1 materials-12-00561-f001:**
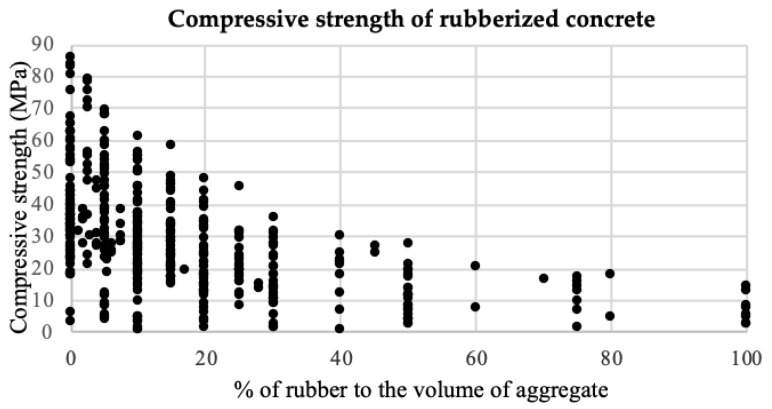
Compressive strength of 457 experiments of rubberized concrete.

**Figure 2 materials-12-00561-f002:**
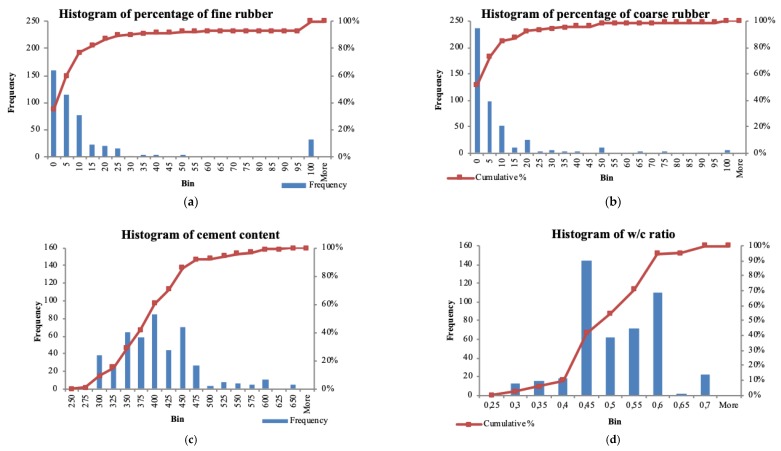
Histogram of: (**a**) percentage of fine rubber, (**b**) percentage of coarse rubber, (**c**) cement ratio, (**d**) w/c ratio.

**Figure 3 materials-12-00561-f003:**
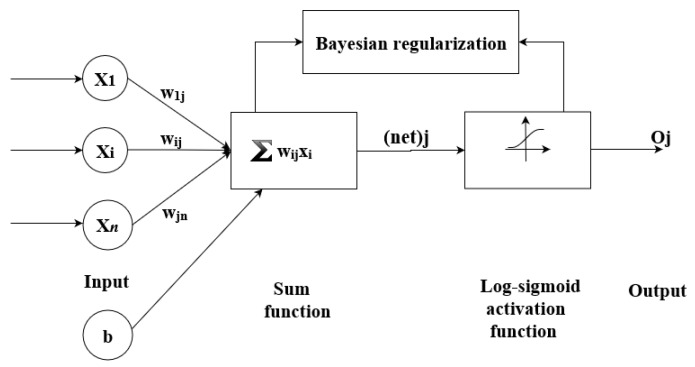
Architecture of applied neural network.

**Figure 4 materials-12-00561-f004:**
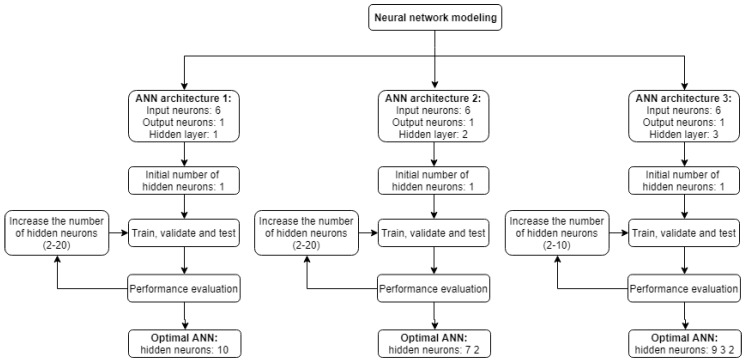
Flow chart for artificial neural network (ANN) model selection.

**Figure 5 materials-12-00561-f005:**
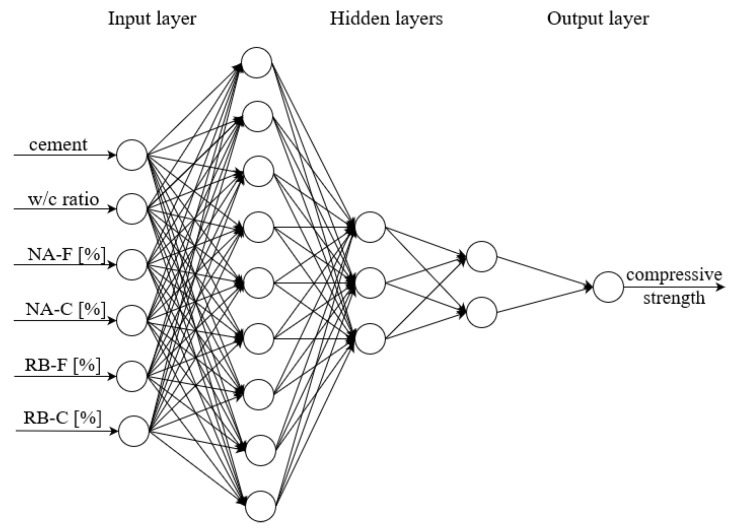
Structure of the best ANN network: three hidden layers with nine, three and two hidden neurons.

**Figure 6 materials-12-00561-f006:**
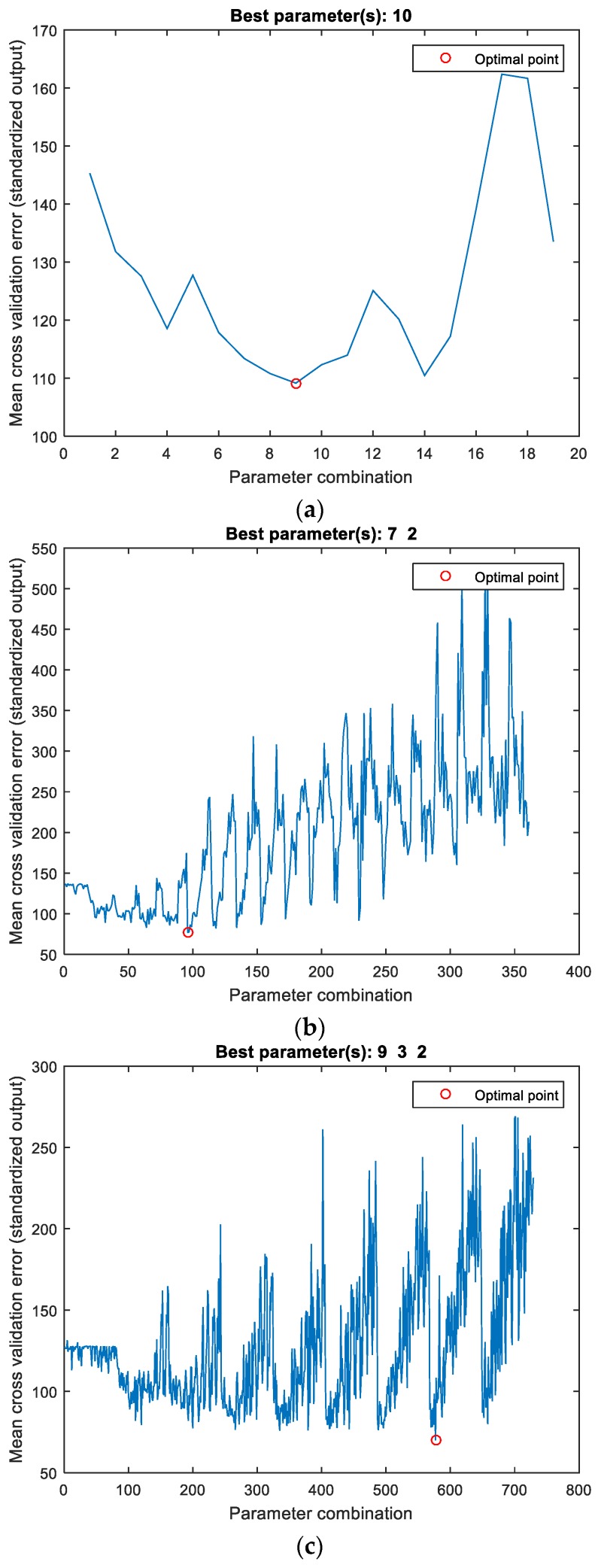
Determination of the best ANN model with (**a**) one hidden layer, (**b**) two hidden layers and (**c**) three hidden layers.

**Figure 7 materials-12-00561-f007:**
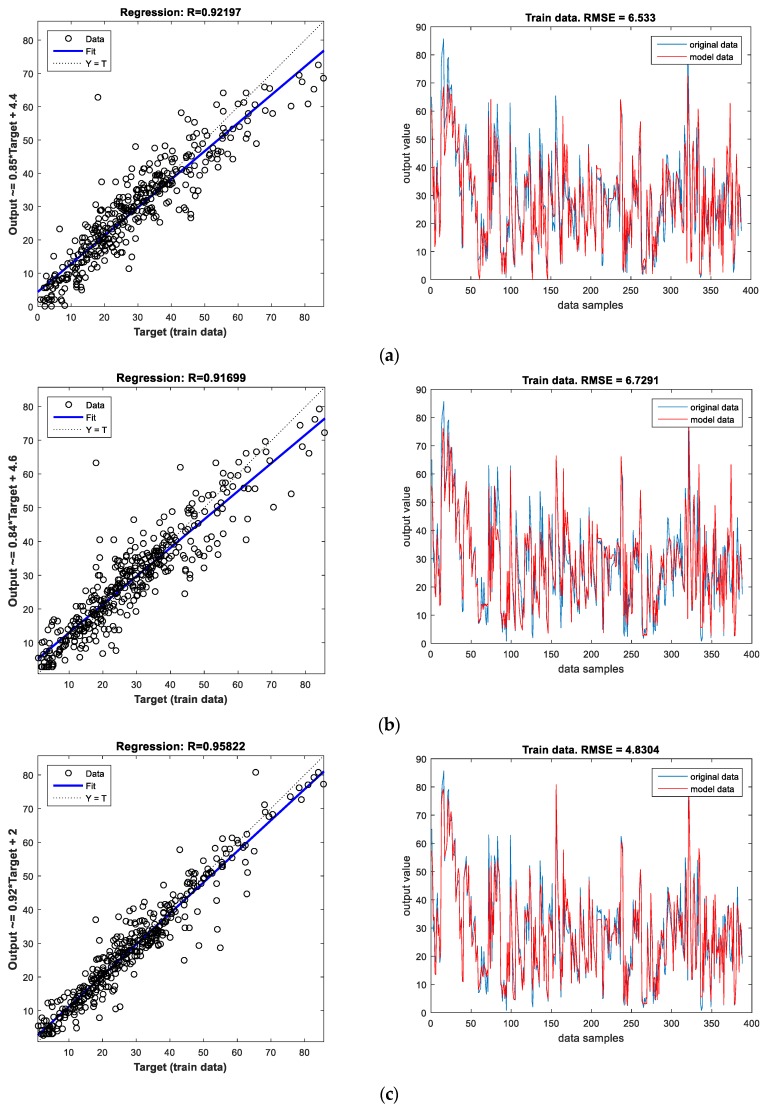
Performance evaluation using the train dataset for ANN models with (**a**) one hidden layer and ten neurons in the hidden layer, (**b**) two hidden layers having seven and two neurons respectively, (**c**) three hidden layers having nine, three and two neurons respectively.

**Figure 8 materials-12-00561-f008:**
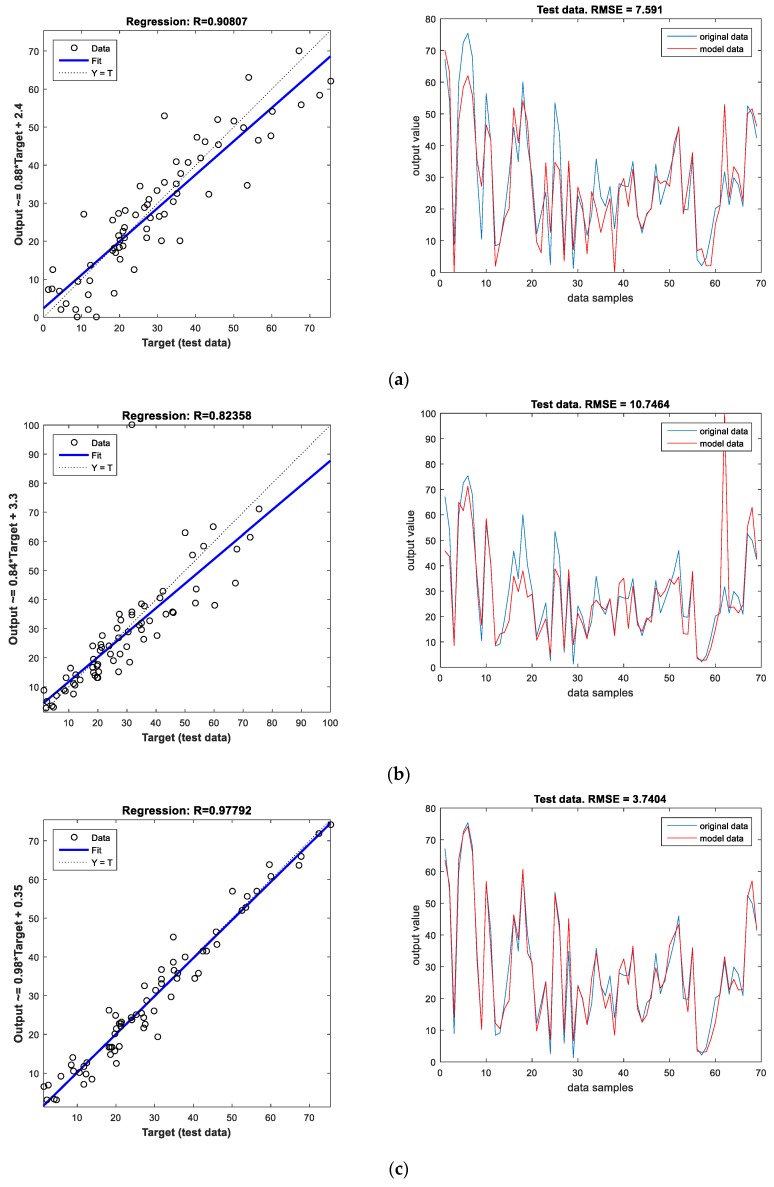
Performance evaluation using the test dataset for ANN models with (**a**) one hidden layer and ten neurons in the hidden layer, (**b**) two hidden layers having seven and two neurons respectively, (**c**) three hidden layers having nine, three and two neurons respectively.

**Figure 9 materials-12-00561-f009:**
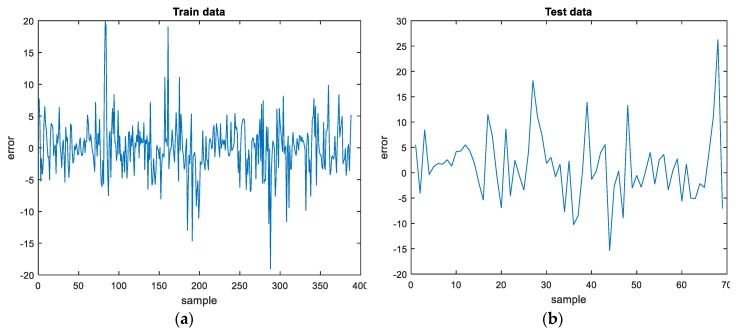
Percentage error for: (**a**) train and (**b**) test data.

**Table 1 materials-12-00561-t001:** Database of concrete with tire rubber.

No.	Author(s)	Ref	Year	Rubber Content (%)	Rubber Size (mm)	No. of Samples
1	Toutanji	[2]	1996	25, 50, 75, 100	Maximum size of 12.7	5
2	Guneyisi et al.	[26]	2004	2, 5, 5, 10, 15, 25, 50	Between 10–40	70
3	Albano et al.	[29]	2005	5, 10	0.29 and 0.59	13
4	Geosglu and Guneyisi	[30]	2007	5, 15, 25	10–40	16
5	Skripkiūnas et al.	[11]	2007	3.2	0–1	2
6	Azmi et al.	[31]	2008	10, 15, 20, 30	Up to 10	15
7	Batayneh et al.	[32]	2008	20, 40, 60, 80, 100	From 0.15 to 4.75	6
8	Taha et al.	[33]	2008	25, 50, 75, 100	From 5–10 and from 10–20	9
9	Turatsinze and Garros	[34]	2008	10, 15, 20, 25	4–10	5
10	Zheng	[35]	2008	15, 30, 45	Up to 2.6	7
11	Ganjian et al.	[27]	2009	5, 7.5, 10	Powder rubber and chipped up to 15	4
12	Aiello and Leuzzi	[36]	2010	15, 25, 30, 50, 75	10–20	9
13	El-Gammal et al.	[13]	2010	50, 100	Maximum size of 19	4
14	Ozbay et al.	[37]	2010	5, 15, 25	0–3	4
15	Paine and Dhir	[6]	2010	2, 4, 6	0.5–1.5, 2–8 and 5–25	13
16	Sgobba et al.	[38]	2010	30	Maximum size: Ash–1, Crumb–14, Chips–25	8
17	Ghedan and Hamza	[39]	2011	15	Maximum size of 10	2
18	Son et al.	[40]	2011	5, 10	0.6 and 1	6
19	Grinys et al.	[41]	2012	5, 10, 20, 30	0–1, 1–2 and 2–3	12
20	Rahman et al.	[42]	2012	28	1–4	4
21	Siringi et al.	[43]	2012	7.5, 10, 17	Up to 4.76	17
22	Al-Tayeb et al.	[44]	2013	5, 10, 20	1	4
23	Dong et al.	[45]	2013	15, 30	Up to 4.75	5
24	Bala et al.	[46]	2014	10, 20, 30, 40	Fine (<4.75mm), coarse (4.75–10 mm) and coarse (10–20 mm)	5
25	Fiore et al.	[47]	2014	10, 20, 30, 40, 50, 75	2–4	7
26	Geosglu et al.	[48]	2014	10, 20	10 and 40	11
27	Kumar et al.	[49]	2014	Powder: 10, 20, 30, 40 Chipped: 2.5	Rubber powder and chipped rubber about 20	12
28	Mohammadi et al.	[50]	2014	10, 20, 30, 40	10 and 20	12
29	Mohammed and Azmi	[28]	2014	10, 15, 20, 30	0.6	45
30	Onuaguluchi and Panesar	[51]	2014	5, 10, 15	Up to 2.3	10
31	Toma et al.	[52]	2014	10	up to 4	2
32	Topličić-Ćurčić et al.	[53]	2014	10, 20, 30	1–4	4
33	Wang and Huang	[54]	2014	10, 20, 30, 40, 50	0.18–0.25	6
34	Youssf et al.	[1]	2014	5, 10, 20	Between 0.15–2.36	12
35	Abusharar	[55]	2015	10	Powder (0.2–0.6), fine dust (0.4–1) and 1	4
36	Herrera-Sosa et al.	[56]	2015	5, 10, 15, 20	0.85 and 2.8	7
37	Ismail et al.	[57]	2015	5, 10, 15, 20, 30, 40	Up to 4.75	7
38	Khan and Singh	[58]	2015	2, 5, 10	4.75–10	4
39	Mishra and Panda	[59]	2015	5, 10, 15, 20	5–10	5
40	Richardson et al.	[60]	2015	0.6	Up to 2.5	6
41	Selvakumar et al.	[61]	2015	5, 10, 15, 20	Fine (<4 mm)	5
42	Toma et al.	[62]	2015	40, 60, 80	Up to 4	4
43	Asutkar et al.	[63]	2016	5, 10, 15, 20	10–20	5
44	Ishwariya	[64]	2016	20	From 0.15 to 4.75	2
45	Liu et al.	[65]	2016	5, 10, 15, 20 1, 3, 5, 10	Maximum size of 31.5	9
46	Sofi	[66]	2016	5, 7.5, 10% by weight	2–4	4
47	Zaoiai et al.	[67]	2016	2.5, 5, 10, 15	Fine rubber (0–3), coarse rubber (3–8)	5
48	Almaleeh et al.	[68]	2017	25, 50, 75, 100	Up to 20	18
49	Bharathi Murugan et al.	[69]	2018	5, 10, 15, 20, 25	Up to 4.75	6
					**Total**	**457**

**Table 2 materials-12-00561-t002:** Simple statistical analysis of the data in the experimental database.

	Parameter	Min	Max	Average
	Cement (kg)	270	629.27	393.15
	w/c ratio	0.27	0.68	0.48
**Input**	NA-F 0–4 (%)	0	100	85.16
	NA-C 4–16 (%)	0	100	91.91
	RB-F 0–4 (%)	0	100	12.43
	RB-C 4–16 (%)	0	100	6.56
**Output**	*f*_c,28_ (MPa)	0.78	85.7	28.65

**Table 3 materials-12-00561-t003:** Statistical performance of best ANN model obtained using cross validation procedure.

ANN model		Statistical Errors					
		*R*	*RMSE*	*MAPE*	*R*	*RMSE*	*MAPE*
No. of Hidden Layers	No. of Neurons	Train Data			Test Data		
1	10	0.9220	6.5330	24.06	0.9081	7.591	41.32
2	7-2	0.9170	6.7291	27.72	0.8236	10.7464	33.46
3	9-3-2	0.9582	4.8304	20.19	0.9779	3.7404	21.64

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
