# Peer review of "Modelling the Influence of Waste Rubber on Compressive Strength of Concrete by Artificial Neural Networks"

_materials, 2019, doi:10.3390/ma12040561_

Reviewer 1 Report

The article is devoted to the application of artificial intelligence for the identification of rubberized concrete. The article is very interesting. However, it needs major revision under the following guidances:

- the text should be carefully edited before acceptance. There are many places with missed information, such as line 41 ("was investigated by..."),

- in the Introduction I suggest to discuss single references when cited. For example it does not much information to the reader that it is written that "in particular [17-22]" in line 71. Maybe some references are reduntant and should be deleted as unappropiate,

- the article is too long. I suggest to reduce the length of the manuscript to be less than 20 pages,

- line 109. What kind of sites and sources were used to search the literature? Was it Scopus or Springer? Please specify,

- Table 1 does not present any experimental database of concrete with tire rubber and there is no  list of authors and samples. Please update the caption of this table,

- Fig. 1. Please specify and mark which point is related to which article from the reference list,

- page 10. Please provide a mathemiathical formula for log-sigmoid activation function,

- line 239. Authors stated that "ANN is found to be an effective tool for engineering applications". Please provide appropiate references to this statement,

- line 262. Please specify how many of data were used for training, validation and testing,

- text between the lines 290-292 is reduntant. It seems that it has been left from the template,

- please provide a more detailed conclusions,

Author Response

We wish to thank the reviewer for accepting our paper and for helping us make the paper better and easier to understand. In this document, we provide answers to all reviewer's comments and explain all modifications made to our paper according to the suggestions of the reviewer.

Reviewer 2 Report

To enhance the quality of analysis results, Figs.9-11 and Figs. 12-14 should be changed to the condensed format.  

Author Response

First, we wish to thank the reviewer for accepting our paper and for helping us make the paper better and easier to understand. In this document, we provide answers to all reviewer's comments and explain all modifications made to our paper according to the suggestions of the reviewer.

We would also like to mention that, while modifying the paper, we realized that due to an error on our part we presented the wrong values and diagrams for the best ANN model with 3 hidden layers (we used a slightly modified dataset when generating the final results for this ANN network structure). These values and diagrams have been corrected in the paper. The values and diagrams are related to the last row in Table 3 and Figures 7c and 8c. These values and diagrams have no impact on the conclusions arrived at in the previous version of the paper and od not change the conclusions also made in this paper.

We have condensed Figs. 9-11 to Fig. 6a-c. However, due to the information displayed, Figs. 12-14 have been condensed to Fig.7 and Fig. 8.

Reviewer 3 Report

The manuscript focus on the application of ANNs to predict the compressive strength of concrete containing waste rubber. For this purpose a large database containing 457 mixtures with partial or full replacement of natural aggregate with waste rubber in concrete was collected from different researchers papers published by other researchers. The paper needs a revision before acceptance. Please find the following coments:

- Line 71. I suggest to add another application of ANNs to predict the mosture content in saline brick walls (Goetzke- Pala et al. A non-destructive method of the evaluation of the moisture in saline brick walls using artificial neural networks. Archives of Civil and Mechanical Engineering, 18(4), 1729-1742.)

- Lines 75-76. It will be beneficial to add and discuss recent ANNs and algorithms used recently for the prediction of the compressive strength of concrete, such as the following papers:

Sadowski et al. (2019). Hybrid ultrasonic-neural prediction of the compressive strength of environmentally friendly concrete screeds with high volume of waste quartz mineral dust. Journal of Cleaner Production, 212, 727-740.

Behnood, A., & Golafshani, E. M. (2018). Predicting the compressive strength of silica fume concrete using hybrid artificial neural network with multi-objective grey wolves. Journal of Cleaner Production, 202, 54-64.

- page. 7. Line 2016. It should be rather figure 2 (no fig. 1),

- I suggest to merge figures from 2 to 5 into one figure 2 when fig. 2a will present compressive strength, fig. 2b will present percentage of RB-C replacement compared to the fine natural aggregate, fig. 2c will present cement content and fig. 2d will present w/c ratio,

- page. 10. The quality of fig. 7 is low. Please replace it with the figure with higher resolution,

- page 11. The quality of fig. 8 is also low. It is a print screen (green marks are visible). Please replace it with the figure with better quality and higher resolution,

- section 4. Please define how many runs of ANN with the same architecture were employed to obtain the average value.

Author Response

Response to reviewer:

The manuscript focus on the application of ANNs to predict the compressive strength of concrete containing waste rubber. For this purpose a large database containing 457 mixtures with partial or full replacement of natural aggregate with waste rubber in concrete was collected from different researchers papers published by other researchers. The paper needs a revision before acceptance. Please find the following coments:

First, we wish to thank the reviewer for accepting our paper and for helping us make the paper better and easier to understand. In this document, we provide answers to all reviewer's comments and explain all modifications made to our paper according to the suggestions of the reviewer.
We would also like to mention that, while modifying the paper, we realized that due to an error on our part we presented the wrong values and diagrams for the best ANN model with 3 hidden layers (we used a slightly modified dataset when generating the final results for this ANN network structure). These values and diagrams have been corrected in the paper. The values and diagrams are related to the last row in Table 3 and Figures 7c and 8c. These values and diagrams have no impact on the conclusions arrived at in the previous version of the paper and od not change the conclusions also made in this paper.

- Line 71. I suggest to add another application of ANNs to predict the mosture content in saline brick walls (Goetzke- Pala et al. A non-destructive method of the evaluation of the moisture in saline brick walls using artificial neural networks. Archives of Civil and Mechanical Engineering18(4), 1729-1742.)

Answer: We have added the reference.

- Lines 75-76. It will be beneficial to add and discuss recent ANNs and algorithms used recently for the prediction of the compressive strength of concrete, such as the following papers:

Sadowski et al. (2019). Hybrid ultrasonic-neural prediction of the compressive strength of environmentally friendly concrete screeds with high volume of waste quartz mineral dust. Journal of Cleaner Production212, 727-740.

Behnood, A., & Golafshani, E. M. (2018). Predicting the compressive strength of silica fume concrete using hybrid artificial neural network with multi-objective grey wolves. Journal of Cleaner Production202, 54-64.

Answer: We have added the reference and comments/discussions.

- page. 7. Line 2016. It should be rather figure 2 (no fig. 1),

Answer: We have corrected the number of the figure

- I suggest to merge figures from 2 to 5 into one figure 2 when fig. 2a will present compressive strength, fig. 2b will present percentage of RB-C replacement compared to the fine natural aggregate, fig. 2c will present cement content and fig. 2d will present w/c ratio,

Answer: We have merged figures 2 to 5 to create one figure 2.

- page. 10. The quality of fig. 7 is low. Please replace it with the figure with higher resolution,

Answer: We have created a new figure with a higher resolution

- page 11. The quality of fig. 8 is also low. It is a print screen (green marks are visible). Please replace it with the figure with better quality and higher resolution,

Answer: We have created a new fig with a higher resolution and better quality

- section 4. Please define how many runs of ANN with the same architecture were employed to obtain the average value.

Answer: We have added the sentence in section 3 (subsection 3.1):” For a given network architecture, the training procedure was repeated 10 times, with random initialization of weights each time, and the best trained network was selected as a representative of the given architecture

Reviewer 4 Report

The manuscript is in good quality, however, the following revisions needs to be made:

(1) Please improve the English proficiency of the manuscript.

(2) Please mention the references for your Figure 4 and Figure 8. 

(3) The literature review needs to be improved with more related research of scientists in recent years. You may want to refer to the following paper:

(a Khademi, F., Akbari, M., Jamal, S. M., & Nikoo, M. (2017). Multiple linear regression, artificial neural network, and fuzzy logic prediction of 28 days compressive strength of concrete. Frontiers of Structural and Civil Engineering11(1), 90-99.

(3) Please explain how you have devided your data into three sets of training, validation, and test?

(4) Please explain what algorithms you have tried for your research, and name the one with the best fitted results for your ANN model? 

(5) The theoretical concept of your ANN model needs to be improved.

(6) The conclusion needs to be specified in more details. 

The manuscript is in good quality, and my decision is minor revision. After these revisions are made, the manuscript has a high chance of getting accepted in this journal.

Author Response

Response to reviewer:

The manuscript is in good quality, however, the following revisions needs to be made:

 First, we wish to thank the reviewer for accepting our paper and for helping us make the paper better and easier to understand. In this document, we provide answers to all reviewer's comments and explain all modifications made to our paper according to the suggestions of the reviewer.

We would also like to mention that, while modifying the paper, we realized that due to an error on our part we presented the wrong values and diagrams for the best ANN model with 3 hidden layers (we used a slightly modified dataset when generating the final results for this ANN network structure). These values and diagrams have been corrected in the paper. The values and diagrams are related to the last row in Table 3 and Figures 7c and 8c. These values and diagrams have no impact on the conclusions arrived at in the previous version of the paper and od not change the conclusions also made in this paper.

(1) Please improve the English proficiency of the manuscript.

 Answer: We have given the paper to a native speaker of English to review the text

(2) Please mention the references for your Figure 4 and Figure 8. 

 Answer: Figures 4 and 8 are our figures and as such have no references

(3) The literature review needs to be improved with more related research of scientists in recent years. You may want to refer to the following paper:

(a Khademi, F., Akbari, M., Jamal, S. M., & Nikoo, M. (2017). Multiple linear regression, artificial neural network, and fuzzy logic prediction of 28 days compressive strength of concrete. Frontiers of Structural and Civil Engineering11(1), 90-99.

 Answer: We have added the reference.

(3) Please explain how you have devided your data into three sets of training, validation, and test?

Answer: We have added the sentence: “In this study, 70% of the original experimental data was randomly selected as the training dataset, 15% as the validation dataset and 15% as the test dataset.”

(4) Please explain what algorithms you have tried for your research, and name the one with the best fitted results for your ANN model? 

Answer: In this study, we only used the the Levenberg-Marquardt method in combination with Bayesian regularization, to determine the optimal values of the weights.

We have added a sentence in the paper mentioning this fact: “In this study, the Levenberg-Marquardt method is used, in combination with Bayesian regularization, to determine the optimal values of the weights. This optimization procedure minimizes a combination of squared errors and weights, and then determines the suitable combination in order to enhance the generalization properties of the network.

“(5) The theoretical concept of your ANN model needs to be improved.

Answer: The whole section 3 has been modified a bit.      

We have added a bit more information about the ANN model such as:

“In this study, the Levenberg-Marquardt method is used, in combination with Bayesian regularization, to determine the optimal values of the weights. This optimization procedure minimizes a combination of squared errors and weights, and then determines the suitable combination in order to enhance the generalization properties of the network. “

“In this study, 70% of the original experimental data was randomly selected as the training dataset, 15% as the validation dataset and 15% as the test dataset”

(6) The conclusion needs to be specified in more details. 

Answer: A more detailed conclusion has been provided.

The manuscript is in good quality, and my decision is minor revision. After these revisions are made, the manuscript has a high chance of getting accepted in this journal.

Round  2

Reviewer 1 Report

The changes bwere made and the paper can ne accepted.